# Prognostic Factors of Renal Outcomes after Heart Transplantation: A Nationwide Retrospective Study

**DOI:** 10.3390/jcm10215110

**Published:** 2021-10-30

**Authors:** Junseok Jeon, Hyejeong Park, Youngha Kim, Danbee Kang, Jung Eun Lee, Wooseong Huh, Eliseo Guallar, Juhee Cho, Hye Ryoun Jang

**Affiliations:** 1Division of Nephrology, Department of Medicine, Sungkyunkwan University School of Medicine, Seoul 06531, Korea; uncleimdr@gmail.com (J.J.); jungeun34.lee@samsung.com (J.E.L.); wooseong.huh@samsung.com (W.H.); 2Center for Clinical Epidemiology, Samsung Medical Center, Sungkyunkwan University School of Medicine, Seoul 06351, Korea; inx1234@sbri.co.kr (H.P.); haha1223@sbri.co.kr (Y.K.); dbee.kang@skku.edu (D.K.); eguallar@jhu.edu (E.G.); 3Department of Epidemiology and Medicine and Welch Center for Prevention, Epidemiology and Clinical Research, Johns Hopkins Medical Institutions, Baltimore, MD 21287, USA

**Keywords:** chronic kidney disease, end-stage kidney disease, heart transplantation, nationwide cohort, renal outcome

## Abstract

Renal dysfunction after heart transplantation (HT) is associated with poor survival. We investigated the predictive factors of renal outcomes after HT using nationwide cohort data. In this retrospective cohort study using the Health Insurance Review and Assessment database of Korea, 654 patients who received HT between 2008 and 2016 and survived until discharge after HT were analyzed. The median (interquartile range) age was 52 (40–60) years, and 68.1% were male. Perioperative renal replacement therapy (RRT) was performed in 27.8% of patients. During 2.8 years of median followup, end-stage kidney disease (ESKD) developed in 12 patients (1.8%). In a fully adjusted model, RRT > 3 weeks, the use of inotropes/vasopressors and non-use of ACEi/ARB were associated with ESKD. Preexisting renal disease tended to be associated with ESKD. Among the 561 patients without preexisting CKD, 104 (18.5%) developed chronic kidney disease (CKD). Age, extracorporeal membrane oxygenation, and RRT were associated with the development of CKD after HT. Our nationwide cohort study demonstrated that perioperative RRT was a predictor of poor renal outcomes after HT. These results suggest that an active renoprotective strategy is required during the perioperative period.

## 1. Introduction

Heart transplantation (HT), which is a definitive treatment for end-stage heart failure, is associated with good long-term survival and improved overall prognosis over time [1,2]. Renal dysfunction is common in patients on the waiting list for HT and is an important prognostic factor for HT [3,4]. The International Society for Heart and Lung Transplantation recommends simultaneous heart and kidney transplantation (HKT) in patients with severe irreversible renal dysfunction, defined as an estimated glomerular filtration rate (eGFR) of <30 mL/min/1.73 m^2^ [5]. However, consensus and criteria regarding the indication for simultaneous HKT are lacking.

Cardiorenal syndrome (CRS) is defined as a disorder in which acute or chronic dysfunction of the heart or kidney induces acute or chronic dysfunction of the other. CRS is a leading cause of renal dysfunction in patients with heart failure. Acute CRS is known to be reversible [6,7]; however, there have been conflicting results regarding the clinical impact of pre-HT renal dysfunction on the overall prognosis of HT [3,8,9,10]. Given the organ shortage and the clinical difficulty of determining the reversibility of renal dysfunction in patients with advanced renal dysfunction being considered for HKT, it is imperative to identify predictive factors of renal outcome after HT. However, few studies have focused on the perioperative factors that predict renal function after HT. In addition, generalization issues regarding the indications of HKT or predictive factors for post-HT renal outcomes are raised because most previous studies were single-center based, and indications or management protocols may differ between centers.

We aimed to analyze the predictive factors for renal outcome after HT using the Korean National Health Insurance Database, a representative data set for all HT recipients in Korea.

## 2. Materials and Methods

### 2.1. Study Population

We conducted a retrospective cohort study using the Korean intensive care unit (ICU) National Data (KIND) registry [11]. The KIND registry included all patients aged >18 years with at least one ICU admission covered by the Korean National Health System between January 2007 and September 2016. South Korea has a single-payer national health system. The Health Insurance Review and Assessment (HIRA) database contains health insurance claims data generated in the process of reimbursing claims for healthcare services under the National Health Insurance system in South Korea. The HIRA database covers 98% of the total population through a universal health coverage system [12]. Thus, the KIND database includes all ICU admissions in South Korea.

All patients aged ≥18 years who underwent HT between January 2008 and May 2016 (codes Q8080, Z94.1, and Z94.3; *n* = 752) were screened. Patients who underwent kidney or liver transplantation before admission (codes Q8040-Q8050, Q8140-Q8150, or Z94.4; *n* = 3) or HKT (codes R3280 or Z94.0; *n* = 13) were excluded.

All HT recipients who survived until discharge (*n* = 654) were included in the analysis of potential risk factors for end-stage kidney disease (ESKD) after discharge. Subjects without a history of preexisting CKD (*n* = 561) before HT were included in the analysis of potential risk factors for chronic kidney disease (CKD) after discharge. Preexisting CKD was defined as presence of as least one relevant diagnostic code (codes N18, N19, I12, I13, E10.2, E11.2, E13.2, and E14.2) within one year before admission.

### 2.2. Measurement

The primary outcome was ESKD development after HT. ESKD was defined as three or more peritoneal dialysis (codes O7020, O7021, O9991, O2011, O2012, and O2081-O2083), ≥40 hemodialysis within 5 months (codes O7061, O7062, O7071-O7075, E6581, E6582, and E6593), or kidney transplantation after discharge. In addition, we also estimated the incidence of CKD, which was defined by at least one relevant diagnostic code (codes N18, N19, I12, I13, E10.2, E11.2, E13.2, and E14.2) after discharge among patients without a prior diagnosis of CKD.

To identify potential risk factors, comorbidities, surgical procedures, and demographic information were included. Comorbidities confirmed within 1 year before admission were summarized using the Charlson’s index as follows: cerebrovascular disease, chronic pulmonary disease, connective tissue disease, liver disease, diabetes mellitus (DM), renal disease, and peripheral vascular disease [13,14]. In addition, hypertension (I10-I13 and I15) was included in the analysis. Mechanical ventilation (M5857, M5858, and M5860) for >3 days, extracorporeal membrane oxygenation (ECMO; O1901-O1904), and renal replacement therapy (RRT), including continuous renal replacement therapy (CRRT; O7051-O7054) and hemodialysis (HD; O7020) were also included in the final analysis. Patients who required RRT were grouped according to the duration of RRT with a cutoff of 21 days. We identified the use of inotropes or vasopressors such as dobutamine, dopamine, epinephrine, norepinephrine, vasopressin, or milrinone administered for >2 days, and angiotensin-converting enzyme inhibitor (ACEi)/ angiotensin II receptor blocker (ARB) prescribed for >2 weeks using the Korean drug and anatomical therapeutic chemical codes (Appendix A) [15]. The length of stay in the ICU was calculated as the submitted number of reimbursements for ICU admission fees (codes AJ100-AJ590). All ICU stays during the same admission were considered as a single ICU admission. Similarly, hospital stays separated by <2 days in the same hospital were considered to be the same hospital admission. Demographic information, including age and sex, was collected according to the information on the first admission date.

### 2.3. Statistical Analyses

Patients were followed up until the development of ESKD or the last available clinic visit. To identify the predictors of ESKD after HT, Cox’s proportional hazards models were used to estimate crude and multivariable-adjusted hazard ratios (HRs) with 95% confidence intervals (CIs). Since mechanical ventilation and ECMO were highly correlated (*p* = 0.691), mechanical ventilation was excluded from the model. Finally, age, sex, comorbidities (cerebrovascular disease, chronic pulmonary disease, liver disease, DM with complications, renal disease, hypertension, and peripheral vascular disease), ECMO, RRT, inotropes, or vasopressors, and ACEi/ARB were included in the model. We estimated HRs using DM with complications instead of all DM to include more clinically relevant variables. Since patients could be clustered by the type of hospital, we used the hospital as a random intercept in the model. CKD incidence after HT was analyzed as the secondary outcome using methods similar to the primary outcome.

The mean and standard deviation or median and interquartile range (IQR) were used to describe the distribution of continuous variables. Chi-squared and Student’s *t*-tests were used to compare categorical and continuous variables, respectively.

All analyses were performed using SAS Enterprise Guide 6.1 (SAS Institute, Cary, NC, USA). *p*-values of <0.05 analyzed with two-sided significance testing were considered statistically significant.

## 3. Results

### 3.1. Baseline Characteristics

Between January 2008 and May 2016, 736 adult patients (aged > 18 years) received HT. The median (IQR) age was 52 (40–60) years, and 68.1% were male (Table 1). Half of the patients had three or more comorbidities. The most frequent comorbidities were hypertension (83.6%), chronic pulmonary disease (47.6%), and DM (41.8%). Out of all the patients, 42.3% required inotropic or vasopressor support, and 27.8% required perioperative RRT during admission. The in-hospital mortality rate was 7.7% (57 patients).

### 3.2. Characteristics According to the Development of CKD after HT

Patients with incident CKD after HT were older (patients without incident CKD vs. patients with incident CKD: 45 (37–58) vs. 49 (53–61), *p* = 0.009) and more frequently males (65.4% vs. 76.0%, *p* = 0.039), with required mechanical ventilation (20.1% vs. 33.7%, *p* = 0.003), ECMO (19.5% vs. 28.8, *p* = 0.035), RRT (16.2% vs. 22.9%, *p* = 0.018), and inotropes or vasopressors (42.9 vs. 30.8%, *p* = 0.023) more frequently than those without incident CKD (Appendix A). ICU length of stay (7 [5,6,7,8,9,10,11,12,13,14,15] vs. 10 [6,7,8,9,10,11,12,13,14,15,16,17,18,19,20], *p* < 0.001) was longer, but hospital length of stay (60 (36–83] vs. 48 (33–73), *p* = 0.039) was shorter in patients with incident CKD than in those without incident CKD.

### 3.3. Factors Associated with the Development of ESKD after HT

A total of 654 patients survived and were discharged alive. During 2,094 person-years of followup (median followup, 2.8 years), ESKD developed in 12 patients (1.8%). The incidence rate of ESKD was 5.7 per 1,000 person-years (Figure 1). In fully adjusted models, perioperative RRT > 21 days (HR 8.64; 95% CI, 3.17–23.17, *p* < 0.001), use of inotropes/vasopressors (HR 6.98; 95% CI, 2.10–23.17, *p* < 0.002), and no use of ACEi/ARB (HR 0.24; 95% CI, 0.08–0.71, *p* < 0.01) were associated with the development of ESKD after HT (Table 2). Preexisting renal disease (HR 3.19; 95% CI, 0.87–2.88, *p* = 0.08) tended to be associated with incident ESKD, but the difference was not statistically significant.

### 3.4. Factors Associated with the Development of CKD after HT

Among 561 patients without preexisting CKD, 104 patients (18.5%) were newly diagnosed with CKD after HT (median followup, 2.3 years). The incidence rate of CKD was 64.9 per 1000 person-years (Figure 2). Old age (HR, 1.03; 95% CI, 1.01–1.05, *p* < 0.01), ECMO (HR, 1.54; 95% CI, 1.14–2.07, *p* < 0.01), and RRT (1–21 days of RRT, HR, 1.76; 95% CI, 1.28–2.41; and >21 days of RRT, HR, 3.69; 95% CI, 1.41–9.68) were associated with an increased risk of newly diagnosed CKD (Table 3). DM with complications tended to be associated with incident CKD, but this was not statistically significant (HR, 1.39; 95% CI, 0.98–1.96, *p* = 0.06).

### 3.5. Characteristics According to Hospital Mortality

Patients who died before discharge had preexisting renal disease (survival vs. death: 15.0% vs. 28.1%, *p* = 0.01) and received more mechanical ventilation (24.6% vs. 86.0%, *p* < 0.001), ECMO (21.6% vs. 61.4%, *p* < 0.001), and RRT (22.5% vs. 91.8%, *p* < 0.001) more frequently than those who survived. ACEi/ARB (32.3% vs. 15.8%, *p* = 0.01) was prescribed less frequently in patients who died after HT than in the surviving patients (Appendix A).

## 4. Discussion

In the present study, we investigated the factors associated with renal outcomes after HT using the Korean National Health Insurance Database between 2008 and 2016. Perioperative RRT was a significant risk factor for poor renal outcomes in a duration-dependent manner. In addition, factors reflecting heart function or general medical conditions, such as ECMO or inotropes/vasopressors, were associated with poor renal outcomes.

The mortality rate of HT is higher in patients with renal dysfunction after HT, especially in dialysis-dependent patients, than in those without renal dysfunction [8,9,10]. The International Society for Heart Lung Transplantation recommends HKT for patients with pre-HT eGFR < 30 mL/min/1.73 m^2^ [5] based on the better overall prognosis of HKT than HT alone in heart failure patients with severe renal dysfunction [3,4,16]. However, it is difficult to determine the reversibility of renal dysfunction in patients with advanced heart failure, and previous studies have not distinguished reversible acute kidney injury (AKI) from CKD in patients with renal dysfunction. In addition, some studies reported that post-HT renal dysfunction was an important predictor of mortality after HT, but pre-HT renal dysfunction was not [9,10]. Therefore, identifying patients at high risk for the development of post-HT ESKD or CKD is an important issue for the more active application of HKT or renoprotective strategies after HT in high risk patients. Previous studies have reported old age, female sex, and DM as factors associated with renal dysfunction after HT [10,17,18,19,20,21].

Our study demonstrated that preexisting renal disease tended to be associated with ESKD after HT. Although the association was not statistically significant, given the relatively short median followup period and low incidence of ESKD, preexisting renal disease is expected to show a significant association with ESKD in a longer followup period. The diagnostic code-based definition of preexisting renal disease in this study may reflect the presence of CKD rather than low preoperative eGFR, including reversible AKI. Therefore, our data suggest that pre-HT CKD may be associated with poor renal outcomes after HT. In addition, patients with preexisting renal disease showed a higher in-hospital mortality rate than those without preexisting renal disease.

Perioperative RRT was associated with the development of both ESKD and CKD after HT. Previous studies have shown that perioperative RRT is associated with mortality; however, its association with post-HT renal function has not been thoroughly evaluated [8,10]. One study reported postoperative RRT as a risk factor for CKD stage 3B or worse after HT, while in another study, postoperative RRT was not associated with ESKD after HT [10,20]. The definition of renal outcome was different; however, these conflicting results may be caused by preoperative renal dysfunction and the severity of the patient’s overall medical condition affecting preoperative or postoperative RRT. We could not distinguish preoperative RRT from postoperative RRT; however, RRT > 3 weeks may appropriately reflect the severity of renal dysfunction and overall medical condition. Further studies are needed to reveal the clinical impact of postoperative AKI or preoperative renal dysfunction on post-HT renal outcomes.

ECMO and the use of inotropes/vasopressors may reflect the severity of heart failure. In our study, inotropes/vasopressors were associated with post-HT ESKD, but ECMO was not. In contrast, the development of post-HT CKD was associated with ECMO, but the use of inotropes or vasopressors was not. In a few studies, ventricular assisted devices or intra-aortic balloon pumps were not associated with renal outcomes after HT, but no study has assessed the effect of perioperative ECMO on renal outcomes after HT [10,20]. Since this study had a relatively short followup period within a median of 3 years, the severity of heart failure may be a major determinant of progression to ESKD. One possible explanation is that patients who were severely ill enough to develop ESKD received more ECMO, so more patients who received ECMO died before reaching ESKD than those who did not receive ECMO. The proportion of ECMO appliance was much higher in patients who died before discharge than in those who survived until discharge, supporting our hypothesis.

In most previous studies, DM was an important risk factor for post-HT renal dysfunction [10,20,22]. In our study, DM with complications tended to be associated with CKD but not with ESKD. Patients with better renal function might have been selected among patients with diabetes compared to non-diabetic patients because of concerns regarding poor outcomes related to diabetic nephropathy.

In our study, the use of ACEi/ARBs during hospitalization was negatively associated with the development of ESKD but not with CKD. A previous study evaluated the effect of ACEi/ARBs on renal function improvement after HT, and no association was found [23]. ACEi/ARBs might not have been prescribed for patients at high risk of ESKD because of severe hypotension or renal dysfunction. This is consistent with the finding that ACEi/ARBs were prescribed less to patients who died during hospitalization.

This study had several limitations. First, as a retrospective study, it is challenging to clarify the causal relationship between risk factors and post-HT renal outcomes and to exclude all potential confounders. However, we tried to minimize bias from regional differences in patients’ socioeconomic status or differences in medical practice, which is a problem in single-center studies, and included major factors that affect renal outcomes. The second limitation is the lack of individual laboratory results and detailed information of drug administration, especially inotropes/vasopressors, due to the inherent limitation of the Korean National Health Insurance database. The claim code for health insurance might underestimate or overestimate the kidney disease burden and cannot stratify CKD according to the Kidney Disease: Improving Global Outcomes (KDIGO) protocol. We tried to overcome this limitation by using ESKD as the endpoint, as this is less likely to be affected by classification errors. More detailed data are required to clarify the clinical significance of these results. Third, the median followup period was relatively short. Many patients have received HT in recent years, resulting in a short overall followup period and a low incidence of ESKD. However, it was possible to identify high risk factors for the development of ESKD within a short period after HT. Fourth, we were not able to analyze the effects of calcineurin inhibitors (CNIs), including cyclosporine and tacrolimus, on renal outcomes due to the fundamental limitation of our insurance reimbursement database. High CNIs levels during early posttransplant period may contribute to kidney failure [24]. However, some studies reported that there was no relationship between the concentration of CNIs and renal function, indicating that many factors in addition to CNIs may affect renal function during early postoperative period [25,26]. A large prospective cohort study is required to reveal the overall effects of CNI on renal function in HT patients. Despite these limitations, our study is still meaningful in that major clinical and perioperative factors affecting post-HT renal outcomes were evaluated in a large cohort of nationwide data.

Our nationwide study demonstrated that preexisting renal disease tended to be associated with poor renal outcomes after HT, and perioperative RRT (especially >3 weeks) was an important predictor of poor renal outcomes after HT. These results suggest that HT should be performed before irreversible renal damage occurs and that an active renoprotective strategy may be required during the perioperative period. Further studies using a large prospective cohort with more detailed data are required to develop a new guideline with more practical renoprotective management and indications of HKT or HT alone.

## Figures and Tables

**Figure 1 jcm-10-05110-f001:**
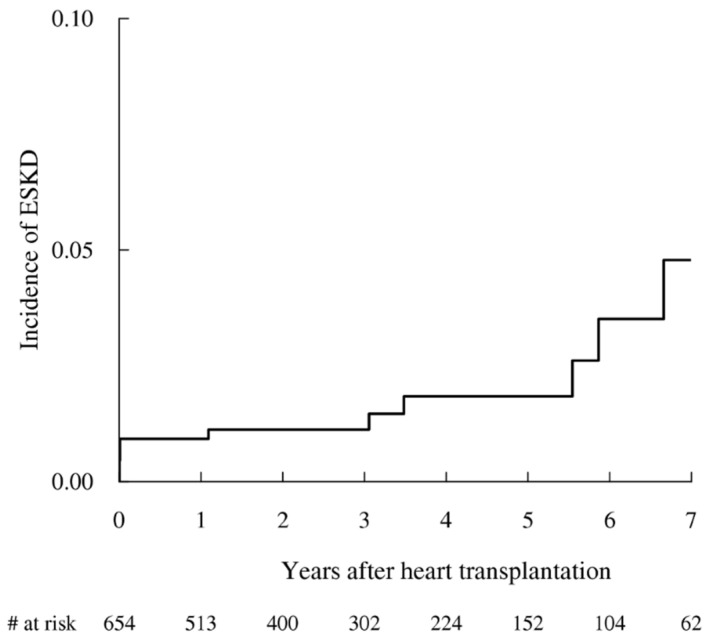
Incidence of end-stage kidney disease (ESKD) after heart transplantation. The median followup duration was 2.8 years (interquartile range: 1.2–4.8), and 12 patients (1.8%) developed ESKD. In the Kaplan-Meier analysis, the incidence rate of ESKD was 5.7 per 1000 person-years.

**Figure 2 jcm-10-05110-f002:**
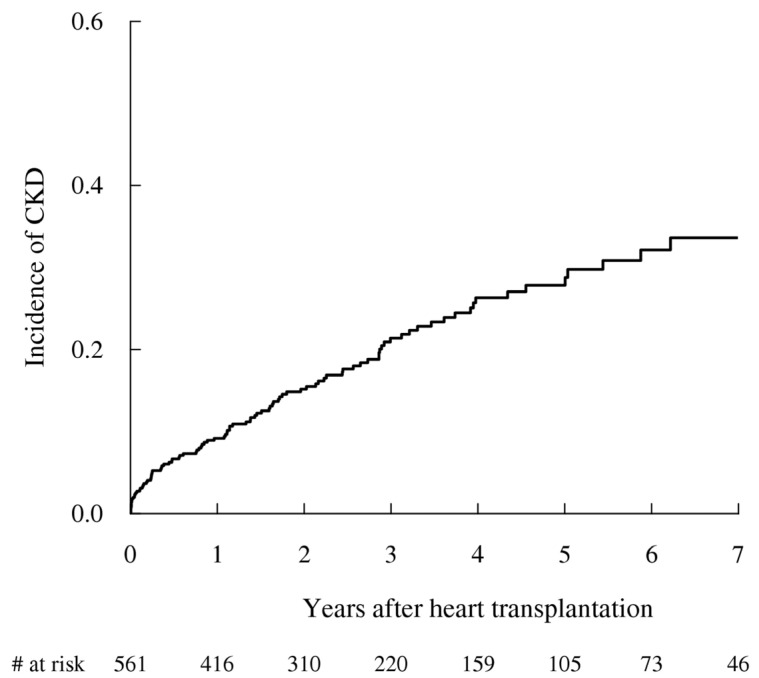
Incidence of chronic kidney disease (CKD) after heart transplantation. The median followup duration was 2.6 years (interquartile range: 1.1–4.6), and 104 patients (18.5%) were newly diagnosed with CKD after heart transplantation. The incidence rate of CKD was 64.9 per 1000 person-years.

**Table 1 jcm-10-05110-t001:** Characteristics of the patients that underwent heart transplantation.

	Overall (*n* = 736)
Age (years)	52 (40–60)
Sex, male	501 (68.1)
Charlson’s index	3 (2–5)
Comorbidity	
Peripheral vascular disease	112 (15.2)
Cerebrovascular disease	114 (15.5)
Chronic pulmonary disease	350 (47.6)
Connective tissue disease	36 (4.9)
Liver disease	280 (38.0)
Diabetes	308 (41.8)
Renal disease	118 (16.0)
Hypertension	615 (83.6)
History of heart transplantation	13 (1.8)
Mechanical ventilation	216 (29.3)
ECMO	182 (24.7)
Renal replacement therapy (CRRT or HD)	
1–21 days	143 (19.4)
>21 days	62 (8.4)
Inotropes or vasopressors	311 (42.3)
ACEi/ARB	228 (31.0)
ICU length of stay (days)	8 (6–20)
Hospital length of stay (days)	58 (36–83)
Hospital mortality	57 (7.7)

Continuous variables are expressed as the median (interquartile range), and categorical variables are expressed as a number (percentage). ECMO, extracorporeal membrane oxygenation; CRRT, continuous renal replacement therapy; HD, hemodialysis; ACEi, angiotensin-converting enzyme inhibitor; ARB, angiotensin II receptor blocker; ICU, intensive care unit.

**Table 2 jcm-10-05110-t002:** Predictors of end-stage kidney disease (ESKD) after heart transplantation (*n* = 654).

	Crude Model HR (95% CI)	*p* Value	Adjusted Model HR (95% CI)	*p* Value
Age (year)	1.02 (0.98–1.05)	0.33	1.00 (0.97–1.04)	0.87
Sex, male				
Male	0.44 (0.15–1.32)	0.14	0.44 (0.13–1.47)	0.18
Female	Reference		Reference	
Charlson’s index	1.19 (1.03–1.36)	0.02		
Comorbidity				
Peripheral vascular disease	1.35 (0.29–6.33)	0.70	1.11 (0.19–6.68)	0.91
Cerebrovascular disease	1.89 (0.45–8.01)	0.39	1.95 (0.47–8.07)	0.36
Chronic pulmonary disease	2.22 (0.66–7.47)	0.20	1.92 (0.66–5.65)	0.23
Liver disease	1.16 (0.32–4.20)	0.82	1.14 (0.45–2.90)	0.78
Diabetes with complications	1.69 (1.05–2.71)	0.03	0.69 (0.25–1.88)	0.47
Renal disease	4.29 (1.71–10.8)	<0.01	3.19 (0.87–11.71)	0.08
Hypertension	0.84 (0.34–2.09)	0.71	0.75 (0.19–2.88)	0.67
Mechanical ventilation	2.74 (0.93–8.09)	0.07		
ECMO	1.56 (0.33–7.36)	0.58	1.18 (0.35–3.94)	0.79
Renal replacement therapy		<0.001		<0.001
No	Reference		Reference	
1–21 days	3.37 (1.88–6.06)		1.70 (0.66–4.41)	
>21 days	15.06 (4.84–46.8)		8.64 (3.17–23.51)	
Inotropes or vasopressors	4.84 (1.84–12.71)	<0.01	6.98 (2.10–23.17)	0.002
ACEi/ARB	0.46 (0.18–1.18)	0.11	0.24 (0.08–0.71)	0.01

HR, hazard ratio; CI, confidence interval; ECMO, extracorporeal membrane oxygenation; ACEi, angiotensin-converting enzyme inhibitor; ARB, angiotensin II receptor blocker.

**Table 3 jcm-10-05110-t003:** Predictors of chronic kidney disease (CKD) after heart transplantation.

	Crude Model		Adjusted Model	
	OR (95% CI)	*p* Value	OR (95% CI)	*p* Value
Age (year)	1.02 (1.01–1.04)	<0.01	1.03 (1.01–1.04)	<0.01
Sex, male	1.55 (0.88–2.73)	0.13	1.61 (0.85–3.05)	0.15
Charlson’s index	1.09 (1.00–1.19)	0.04		
Comorbidity				
Peripheral vascular disease	0.85 (0.43–1.69)	0.65	0.74 (0.43–1.29)	0.28
Cerebrovascular disease	1.23 (0.43–3.56)	0.70	1.24 (0.37–4.20)	0.73
Chronic pulmonary disease	1.03 (0.65–1.63)	0.90	0.93 (0.59–1.46)	0.74
Connective tissue disease	1.04 (0.03–3.69)	0.95	1.12 (0.26–4.93)	0.88
Liver disease	1.10 (0.83–1.47)	0.51	0.94 (0.68–1.31)	0.71
Diabetes with complications	1.80 (1.29–2.50)	<0.001	1.39 (0.98–1.96)	0.06
Hypertension	0.74 (0.35–1.56)	0.43	0.80 (0.39–1.64)	0.55
History of transplantation	0.50 (0.15–1.64)	0.25	0.51 (0.08–3.08)	0.46
Mechanical ventilation	2.28 (1.53–3.39)	<0.001		
ECMO	2.03 (1.31–3.15)	<0.01	1.53 (1.17–2.00)	<0.01
Renal replacement therapy		<0.001		<0.001
1–21 days	1.97 (1.36–2.88)		1.76 (1.28–2.41)	
>21 days	4.54 (2.01–10.23)		3.69 (1.41–9.68)	
Inotropes or vasopressors	0.96 (0.67–1.38)	0.81	0.95 (0.64–1.42)	0.81
ACEi/ARB	0.95 (0.76–1.18)	0.64	1.10 (0.77–1.58)	0.59

ECMO, extracorporeal membrane oxygenation; ACEi, angiotensin-converting enzyme inhibitor; ARB, angiotensin II receptor blocker.

## Data Availability

The data underlying this article were provided by HIRA with permission. These data are not publicly available due to personal information protection.

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
