# Peer review of "Prognostic Factors of Renal Outcomes after Heart Transplantation: A Nationwide Retrospective Study"

_jcm, 2021, doi:10.3390/jcm10215110_

Round 1

Reviewer 1 Report

In this paper, authors aimed to investigate the predictive factors of renal outcomes after heart transplant with a retrospective cohort study using the Health Insurance Review and Assessment database of Korea. They analyzed data of 654 heart transplant patients between 2008 and 2016. Some comments:

  • how do the authors defined "Subjects without a history of pre-existing renal disease?" Based on creatinin levels? They didn't specify this data in the Methods section;
  • why was ESKD defined as ≥40 hemodialysis within 5 months? 
  • the results are interesting, however I think that we can't analyze renal dysfunction in heart transplants recipients without their immunosoppressive regimen. It is already known that cyclosporine and tacrolimus (first immunosoppressive drugs choice) are consistent with worsening in renal function after heart transplant. In my opinion, authors have to clarify this point adding this primary information.

Minor corrections:

  • Please correct in Table 2: "Table 2. This is a table. Tables should be placed in the main text near to the first time they are cited" and add the title of the Table.
  • I wonder that the following text in Materials and Methods is inappropriate: "Materials and Methods should be described with sufficient details to allow others to replicate and build on the published results. Please note that the publication of your manuscript implicates that you must make all materials, data, computer code, and protocols associated with the publication [...] ethical approval, must list the authority that provided approval and the corresponding ethical approval code". 

Author Response

We deeply appreciate the constructive comments by the reviewers and editorial team. The followings are the detailed response and the revised paragraphs in the track changes version of manuscript.

1. How do the authors defined "Subjects without a history of pre-existing renal disease?" Based on creatinine levels? They didn't specify this data in the Methods section.

- We appreciate your constructive comment. The Korean National Health Insurance database do not contain lab results such as serum creatinine levels. Therefore, we inevitably defined renal disease using diagnostic codes. Preexisting renal disease was defined in patients with at least one relevant diagnostic code (codes N18, N19, I12, I13, E10.2, E11.2, E13.2, and E14.2), and these codes were used to define post-transplant CKD as a major outcome. In order to avoid misunderstanding and highlight cases of de-novo CKD in patients without CKD before heart transplantation, “preexisting renal disease” was revised as “preexisting CKD” as follows (line 74 of the materials and methods section of tract changes version).

“Preexisting CKD was defined as presence of as least one relevant diagnostic code (codes N18, N19, I12, I13, E10.2, E11.2, E13.2, and E14.2) within one year before admission.”

2. Why was ESKD defined as ≥40 hemodialysis within 5 months?

-  We appreciate your considerate comment. We attempted to define ESKD as maintaining dialysis for at least 3 months in order to exclude AKI cases undergoing temporary dialysis. Maintenance hemodialysis is performed three times a week in most ESKD patients, but twice a week in some ESKD patients whose residual renal function remains. If dialysis is performed 3 times a week, approximately 39 dialysis sessions are performed for 3 months, whereas if dialysis is performed twice a week, approximately 26 dialysis sessions are performed for 3 months. Therefore, the total numbers of dialysis for ESKD definition were assumed as 40 or more to include all ESKD patients who had been dialyzed 3 times a week. Instead, the period of hemodialysis was determined as 5 months because it takes 5 months for patients receiving dialysis twice a week to reach more than 40 dialysis sessions.

3. The results are interesting, however I think that we can't analyze renal dysfunction in heart transplants recipients without their immunosuppressive regimen. It is already known that cyclosporine and tacrolimus (first immunosuppressive drugs choice) are consistent with worsening in renal function after heart transplant. In my opinion, authors have to clarify this point adding this primary information.

- We deeply appreciate your constructive comment. Calcineurin inhibitors (CNIs), including tacrolimus and cyclosporin, are the most commonly used immunosuppressants after heart transplantation. The nephrotoxicity of CNI is well known and high CNI levels during early post-transplant period may contribute kidney failure, although some studies reported no relationship between CsA concentration and changes in renal function. We totally agree with your opinion pointing out the absolute necessity of analysis regarding overall CNI effects on renal outcome. However, we could not analyze CNI effects on renal outcome because detailed information regarding drugs such as CNI trough concentration was not available in our cohort due to fundamental limitation of the insurance reimbursement database. We acknowledged and added this limitation in the discussion section as follows (line 283 of the discussion section of tract changes version).

 “Fourth, we were not able to analyze the effects of calcineurin inhibitors (CNIs), including cyclosporine and tacrolimus, on renal outcomes due to fundamental limitation of our insurance reimbursement database. High CNIs levels during early post-transplant period may contribute kidney failure [24]. However, some studies reported that there is no relationship between the concentration of CNIs and renal function, indicating that many factors in addition to CNIs may affect renal function during early postoperative period [25,26]. A large prospective cohort study is required to reveal overall effects of CNI on renal function in HT patients.”

Minor corrections:

4. Please correct in Table 2: "Table 2. This is a table. Tables should be placed in the main text near to the first time they are cited" and add the title of the Table.

- Thank you. Table 2 was revised by adding the title as follows (line 174 of the results section of tract changes version).

Table 2. Predictors of end-stage kidney disease (ESKD) after heart transplantation (N = 654)”

5. I wonder that the following text in Materials and Methods is inappropriate: "Materials and Methods should be described with sufficient details to allow others to replicate and build on the published results. Please note that the publication of your manuscript implicates that you must make all materials, data, computer code, and protocols associated with the publication [...] ethical approval, must list the authority that provided approval and the corresponding ethical approval code".

- We appreciate your considerate review. The sentences that you pointed out were deleted since this part is included in the template provided by the journal.

Reviewer 2 Report

Use of inotrope /pressor is unclear as post HT patients are usually on inotrope/pressors. Is there a duration of time that was associated with CKD or ESKD ?

Authors did analyze the effect of immunosuppression(FK506/Cyclocsporine) use and duration post HT ( 6 months or 1 year) was predictive for CKD of ESKD 

Author Response

We deeply appreciate the constructive comments by the reviewers and editorial team. The followings are the detailed response and the revised paragraphs in the track changes version of manuscript.

1. Use of inotrope/pressor is unclear as post HT patients are usually on inotrope/pressors. Is there a duration of time that was associated with CKD or ESKD ?

- We deeply appreciate your constructive comment. Because inotropes/vasopressors are usually administered immediately after HT in many HT patients, inotropes/vasopressors use was defined as administration of inotropes/vasopressors for more than 2 days. Although the total duration of administration during hospitalization was available, it was not possible to discriminate administration timing (preoperative or postoperative) and method (dose or infusion way such as continuous and intermittent) due to the characteristics of our insurance reimbursement database. We tried to perform further analysis regarding effects of inotropes/vasopressors’ duration on renal outcome, however we do not currently have access to the original data and a new approval for data access requires at least several months. Although further analysis cannot be performed, we acknowledged and added this limitation in the discussion section as follows (line 274 of the discussion section of tract changes version).

“The second limitation is the lack of individual laboratory results and detailed information of drug administration, especially inotropes/vasopressors, due to the inherent limitation of the Korean National Health Insurance database.”

2. Authors did analyze the effect of immunosuppression(FK506/Cyclosporine) use and duration post HT ( 6 months or 1 year) was predictive for CKD of ESKD

- We deeply appreciate your constructive comment. Calcineurin inhibitors (CNIs), including tacrolimus and cyclosporin, are the most commonly used immunosuppressants after heart transplantation. The nephrotoxicity of CNI is well known and high CNI levels during early post-transplant period may contribute kidney failure, although some studies reported no relationship between CsA concentration and changes in renal function. We totally agree with your opinion pointing out the absolute necessity of analysis regarding overall CNI effects on renal outcome. However, we could not analyze CNI effects on renal outcome because detailed information regarding drugs such as CNI trough concentration was not available in our cohort due to fundamental limitation of the insurance reimbursement database. We acknowledged and added this limitation in the discussion section as follows (line 283 of the discussion section of tract changes version).

 “Fourth, we were not able to analyze the effects of calcineurin inhibitors (CNIs), including cyclosporine and tacrolimus, on renal outcomes due to fundamental limitation of our insurance reimbursement database. High CNIs levels during early post-transplant period may contribute kidney failure [24]. However, some studies reported that there is no relationship between the concentration of CNIs and renal function, indicating that many factors in addition to CNIs may affect renal function during early postoperative period [25,26]. A large prospective cohort study is required to reveal overall effects of CNI on renal function in HT patients.”

Reviewer 3 Report

Lines 121-135 are important for authors, but not for readers. It should be written where are the described materials, but not the rules of preparing manuscript

The limitation of this study in second endpoint is lack of stratifying CKD according KDIGO protocol. 

Author Response

We deeply appreciate the constructive comments by the reviewers and editorial team. The followings are the detailed response and the revised paragraphs in the track changes version of manuscript.

1. Lines 121-135 are important for authors, but not for readers. It should be written where are the described materials, but not the rules of preparing manuscript

-  We appreciate your considerate review. The sentences that you pointed out were deleted since this part is included in the template provided by the journal

2. The limitation of this study in second endpoint is lack of stratifying CKD according KDIGO protocol. 

- We totally agree with the reviewer's opinion. The underlined part below was added to the limitation part of the discussion section (line 276 of tract changes version).

 “The second limitation is the lack of individual laboratory results and detailed information of drug administration especially inotropes/vasopressors due to the inherent limitation of the Korean National Health Insurance database. The claim code for health insurance might underestimate or overestimate the kidney disease burden, and cannot stratify CKD according to Kidney Disease: Improving Global Outcomes (KDIGO) protocol.”